# Improving traffic accident severity prediction using MobileNet transfer learning model and SHAP XAI technique

**Omar Ibrahim Aboulola***

College of Computer Science and Engineering, University of Jeddah, Jeddah, Saudi Arabia

* Oaboulola@uj.edu.sa

**Data Availability Statement:** Publicly available online and link is added as reference in the paper. Direct Links: 1.https://www.kaggle.com/datasets/neonninja/nzta-crash-analysis-system-cas 2.

## Abstract

Traffic accidents remain a leading cause of fatalities, injuries, and significant disruptions on highways. Comprehending the contributing factors to these occurrences is paramount in enhancing safety on road networks. Recent studies have demonstrated the utility of predictive modeling in gaining insights into the factors that precipitate accidents. However, there has been a dearth of focus on explaining the inner workings of complex machine learning and deep learning models and the manner in which various features influence accident prediction models. As a result, there is a risk that these models may be seen as black boxes, and their findings may not be fully trusted by stakeholders. The main objective of this study is to create predictive models using various transfer learning techniques and to provide insights into the most impactful factors using Shapley values. To predict the severity of injuries in accidents, Multilayer Perceptron (MLP), Convolutional Neural Network (CNN), Long Short-Term Memory (LSTM), Residual Networks (ResNet), EfficientNetB4, InceptionV3, Extreme Inception (Xception), and MobileNet are employed. Among the models, the MobileNet showed the highest results with 98.17% accuracy. Additionally, by understanding how different features affect accident prediction models, researchers can gain a deeper understanding of the factors that contribute to accidents and develop more effective interventions to prevent them.

## Introduction

The number of automobiles on the road is increasing drastically due to the rapid growth of today's society. Traffic accidents have also grown, resulting in massive human and economic costs [1]. Every year, a substantial number of people are injured or killed in car accidents across the world, resulting in huge human and financial losses. To successfully reduce deaths and damages caused by road traffic accidents (RTAs), it is critical to understand the causes of such incidents as well as the severity of the injuries. The increasing complexity of road infrastructure, along with the growing number of cars on the road, necessitates a data-driven approach to studying accident trends and identifying possible risk factors. It is critical to continue investigating and understanding the causes of RTAs, as well as applying effective ways to limit their occurrence and severity [2, 3].

https://www.nzta.govt.nz/safety/partners/crash-analysis-system/.

**Funding:** The author(s) received no specific funding for this work.

According to a recent World Health Organization (WHO) study on global road safety, traffic accidents are responsible for more than 1.19 million deaths each year and automobile accidents are the leading cause of mortality among young people and teens [4, 5]. The severity of traffic accidents is a significant indicator of traffic accident injury. There are a variety of elements that contribute to traffic accidents of varying severity [6, 7]. In the last 20 years, no substantial reduction in traffic accident fatalities and injuries has been observed. Predictive models can help researchers proactively address accident factors, potentially reducing fatalities, saving costs, and enhancing understanding. The authors discussed weather conditions on different types of roads [8, 9]. Other important factors include lighting conditions, first road class and number, and number of vehicles [10].

The central goal of accident data analysis is to identify the key factors that influence the occurrence of road traffic accidents, ultimately addressing critical road safety issues. The effectiveness of accident prevention strategies predominantly relies on the authenticity of the gathered and estimated data and the suitability of the chosen analysis methods [11, 12]. Choosing the appropriate data analysis method is crucial for revealing the causes of accidents in specific zones or study locations and for reasonably accurately predicting the likelihood of daily accident occurrences or assessing the safety levels for different groups of road users in that area [13]. Consequently, the quality of the research relies on the selection of suitable methods. Machine learning approaches have been employed by authors to predict traffic accidents [14, 15]. Zhang et al. [16, 17] used the generalized random forest to estimate heterogeneous treatment effects in road safety studies, providing local authorities and policymakers with more complete information and improving the efficacy of speed camera programs. Some researchers applied statistical methods [18, 19], reinforcement learning approaches [20, 21], hybrid models [22, 23] and deep learning models [24]. A deep convolutional neural network and random forest are employed for the accident risk prediction method in [25, 26].

Many researchers have attempted to investigate accident-contributing elements; however, little work has been given to explaining black box models [27, 28]. The authors applied five machine learning models and explainable machine learning [29, 30]. The primary goal of this research is to develop an accident injury severity prediction model based on a transfer learning approach and to identify major contributing elements utilizing an explainable approach. The US accident dataset (2016-2021) is utilized for predicting traffic accident severity. This study aims to develop an automated system for categorizing accident severity. In brief, this study makes the following noteworthy contributions:

- A MobileNet model based on transfer learning is employed, showcasing exceptional accuracy in the prediction of road traffic accident severity.

- Experiments are conducted on three deep learning models (Multilayer Perceptron (MLP), Convolutional Neural Network (CNN), and Long Short-Term Memory (LSTM)) and five transfer learning models (ResNET, EfficientNetB4, InceptionV3, Xception, and MobileNet).

- The significance of various features is demonstrated through the utilization of the SHapley Additive exPlanations (SHAP) model.

- The proposed model is also tested on another dataset to prove the generalizability of the model.

The structure of this study is as follows: Section 'Related Work' offers an overview of prior research in this field. Section 'Dataset & Methodology' introduces the proposed approach and describes the deep learning and transfer learning models. Section 'Results and Discussion'

presents the assessment of the proposed approach, including experimental results and related discussions. Lastly, the Section 'Conclusion' serves as the conclusion for this study.

## Related Work

Machine learning has gained popularity in forecasting accident severity in recent years due to its capacity to uncover hidden connections and produce more accurate findings than traditional statistical approaches. Traditional statistical approaches for predicting accident severity include disadvantages such as low accuracy and unrealistic assumptions. Machine learning and deep learning approaches have been used by researchers to improve the effectiveness of the prediction tool. This section offers an overview of some of the prior methodologies used to forecast the severity of traffic accidents.

In terms of traffic accident characteristics, Gan et al. [31] used a random forest method to identify eight traffic accident data attributes to predict the degree of traffic accident validation. Engine capacity, hour of day, vehicle age, month of year, day of week, age range of drivers, vehicle movement, and speed restriction are all factors to consider. The Light-GBM model was 87% accurate. In the second research [32], the authors assessed the efficiency of several machine learning models such as Naive Bayes (NB), Random Forest (RF), adaptive boosting (ADA), and Logistic Regression (LR) in predicting injury severity for road accidents. The RF model had the greatest accuracy rate of 75.5%. Bharti et al. [33, 34] applied deep learning models to predict traffic flow. Yadav and Redhu [35] presented an enhanced car following model by analyzing traffic density and jam.

In Saudi Arabia, Aldhari et al. [36] proposed a machine learning-based approach for predicting the severity of road accidents. The system used three machine learning models, RF, LR, and XG-Boost, and used SHAP to solve bias concerns. Experiments are carried out in two modes: binary class classification and multi-class classification. In the first case, XG-Boost had the greatest accuracy score of 71%, while in the second situation, XG-Boost had the best accuracy score of 94%. Sameen and Pradhan [37] suggested a method for forecasting the severity of two accidents using deep learning models such as multi-layer perceptron (MLP), Bayesian linear regression (BLR), and recurrent neural network (RNN). According to research 5, the RNN model obtained an accuracy of 71.77%.

A basic CART model was suggested in the study [38, 39] to predict the severity of motorcycle accidents. In addition, the Partial Decision Trees (PART) and MLP models were used in the research. The relevant elements associated with the severity of motorcycle collision injuries were also discovered. According to the data, the CART model had an accuracy score of 73.81%, while the PART model had a score of 73.45%. Lin et al [40] suggested a deep learning-based system for traffic accident prediction for the Internet of Vehicles. The authors employed learning models such as DNN, DT C4.5, NB, deep belief network (DBN), MLP, and Bayesian network to predict accident risk. The study's findings revealed that the DNN outperformed other models and performed well for stage one and stage two clustering.

Jamal et al. [41] introduced a network that uses a variety of machine learning models, including RF, LR, DE, and XGBoost, to increase prediction accuracy of road accident severity. The authors discovered that the XGBoost model outperformed other models in terms of individual class accuracy and overall prediction performance. Furthermore, the authors discovered particular elements that have a substantial influence on the severity of traffic accidents using feature importance analysis. The suggested XGBoost model scored an outstanding 93% accuracy. To discover the relevant elements for road accident severity, the author [42, 43] proposed RFCNN, an ensemble learning model that integrates machine learning and deep learning. According to their study, the proposed RFCNN model has achieved a good accuracy score on

**Table 1. An overview of the prior studies.**

| Ref | Methods | Key findings | Limitations |
|---|---|---|---|
| [31] | RF, Light-GBM | Identified 8 attributes for accident validation. | Data limitations, Small dataset |
| [32] | NB, RF, ADA, LR | RF model had the highest accuracy rate of 75.5%. | Limited comparison |
| [36] | RF, LR, XG-Boost, SHAP | Machine learning approach in Saudi Arabia, XG-Boost had the best accuracy. | Limited discussion of results |
| [37] | MLP, BLR, RNN | RNN model achieved an accuracy of 71.77%. | Nor handling of Imbalanced dataset. |
| [38] | CART, PART, MLP | 73.45% (PART) Basic CART model for motorcycle accident severity prediction | Lack of feature importance analysis |
| [40] | DNN, DT C4.5, NB, DBN, MLP, Bayesian Network | DNN outperformed other models in predicting accident risk. | Limited explanation of model selection |
| [41] | RF, LR, DE, XGBoost 93% | XGBoost had the best accuracy and identified influential elements. | Limited model comparison, no feature importance analysis |
| [42] | RFCNN | An ensemble learning model, achieved a high accuracy score. | Feature selection may lead towards overfitting. |
| [44] | ID3, NB, J48, CART | J48 machine learning model achieved 96% accuracy. | Limited model comparison |
| [27] | DNN | THe propose model predict accidental severity with explanation. | Limited models used for comparison. |
| [29] | DT, NB, MLP, SVM, NN | ANN-MLP achieved 76.90% of accuracy | Low accuracy results |

the 20 most relevant characteristics. Bahiru et al. [44] examined the performance of numerous machine learning methods, including ID3, NB, J48, and CART. The accuracy of the J48 machine learning model was found to be 96% in the research.

Cicek et al. [29] applied several machine learning models with explanations to predict the severity of accidents. Authors applied deep learning for multitasking and predicted severity levels of traffic accidents with explanations [27]. They performed experiments on a Chinese dataset for traffic accidents. Existing literature indicates that many researchers have employed machine learning and deep learning to predict the severity of traffic accidents. However, a limited number of studies have conducted comparative analyses of the performance of various deep-learning methods. Furthermore, very little research has investigated the exploration of contributing factors using explanations. Explanation of models enhances transparency, interpretability, explanatory capacity, domain knowledge integration, and scientific coherence of models [45]. This is particularly significant because the majority of prediction methods are commonly regarded as black boxes. Therefore, this study compares five distinct transfer learning methods to investigate their respective predictive capabilities. In a novel contribution, an explainable technique with the proposed model is applied to forecast the most influential factors contributing to accidents in the proposed models. A summary of prior studies is presented in Table 1.

## Dataset & methodology

The dataset used, the deep learning models and transfer learning models used, as well as the parameters for evaluating the performance of these models for the prediction of the severity of traffic accidents, are all covered in detail in this section of the study. The framework adopted in the experiment is presented in Fig 1.

### Dataset

This research makes use of accident data records spanning five years (2016–2020) from New Zealand, which were obtained from the Crash Analysis System (CAS) maintained by the Te Manatu Waka Ministry of Transport. The dataset is also accessible through the open data

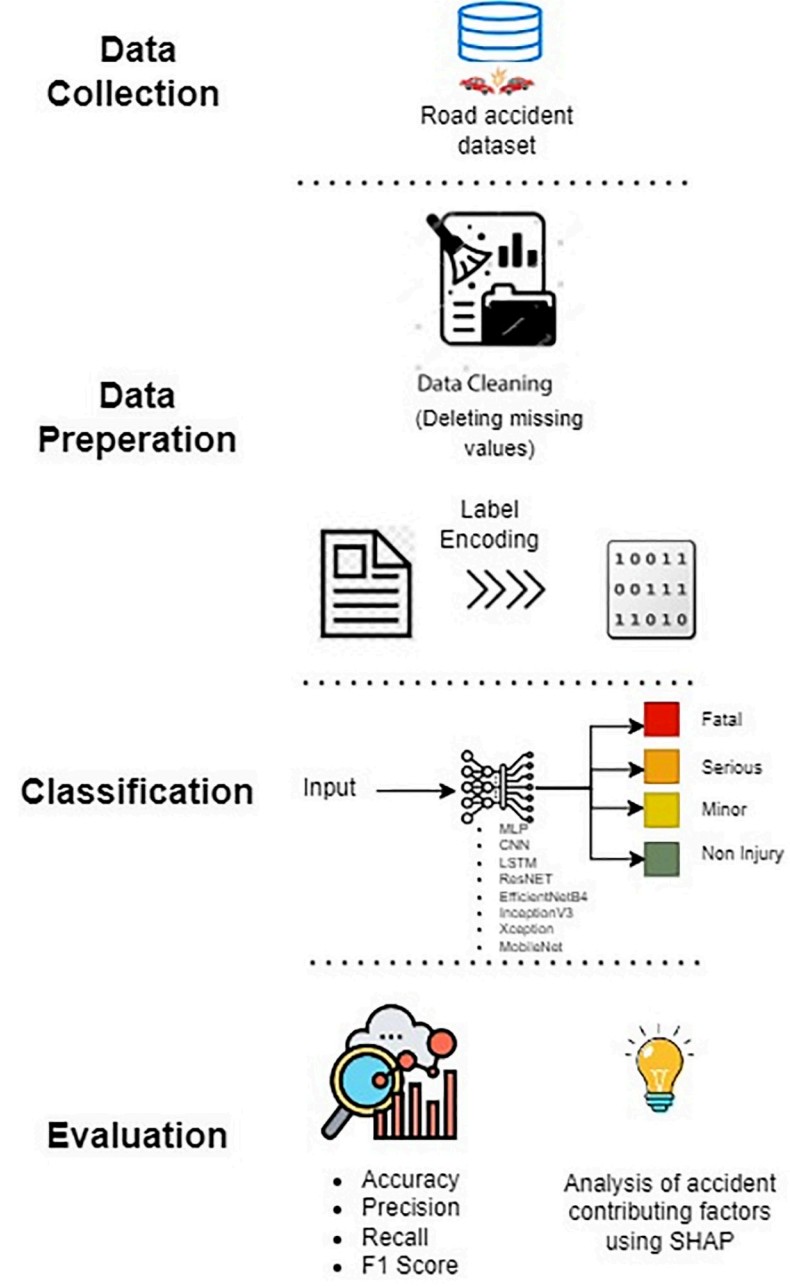

**Fig 1. Architecture of the proposed framework.**

portal. Two data sets were acquired from the CAS system, encompassing information about individuals involved, vehicles, and accident details. These two datasets, known as the 'person' dataset and the 'accident' dataset, were merged to create a comprehensive dataset focusing on factors contributing to accidents. Initially, the combined dataset contained 378,820 rows and 101 columns. However, several columns, out of the 101, were excluded from the study due to their lack of relevance to accident-causing factors. For example, a column containing information about nearby police stations was deemed unnecessary for this research. Consequently, 36 features related to various aspects of accidents are selected. These encompass crash type, crash

**Table 2. Dataset description.**

| Sr. No. | Class | Description | No. of records |
|---|---|---|---|
| 1 | Fatal Crash | A road accident resulting in loss of life. | 1543 |
| 2 | Serious Injury Crash | A road accident in which one or more parties required medical attention and were transported to a hospital. | 10582 |
| 3 | Minor Injury Crash | A road accident in which no one required medical attention but sustained minor injuries like bruises and superficial cuts. | 42888 |
| 4 | Non-Injury Crash | A road accident in which no injuries were sustained, and the presence of law enforcement may not always be required. | 129304 |

location characteristics, environmental factors, vehicle types, vehicle-related factors, and personal influences affecting accident severity. This study categorizes accident types based on their severity. There are four severity levels, defined in Table 2.

## Multilayer Perceptron (MLP)

The multilayer perceptron model [46] is a significant improvement over the original perceptron model by Rosenblatt. While the perceptron was limited to handling linearly separable problems in basic logic, the multilayer perceptron introduces multiple layers of functional neurons, making it capable of addressing nonlinear separable problems. The architecture consists of fully interconnected layers, allowing for the organized flow of information. It uses the error back-propagation algorithm to train, minimizing the cumulative error on the training set, typically measured using mean-square error (MSE) for each sample.

## Convolutional Neural Network (CNN)

CNN [47] is a deep neural network designed for image recognition, classification, and segmentation. It employs convolution, non-linear activation, and pooling layers to extract features. Stacked CNNs are used for specific tasks, such as detecting parasites in infected cell images. CNN's architecture is multi-layered, with each layer applying filters or kernels to input data to create feature maps. The output of convolutional layers is concatenated and fed into fully connected layers for further analysis. CNN has become a standard in medical domain classification and employs secure interaction protocols for privacy-preserving feature extraction.

## Long Short-Term Memory (LSTM)

Long Short-Term Memory (LSTM) [48] is a specialized recurrent neural network (RNN) architecture developed to address the limitations of traditional RNNs in handling long-term dependencies within sequential data. LSTMs are particularly well-suited for a variety of tasks involving sequences, including natural language processing, time series analysis, speech recognition, and more. Key features of LSTMs include their capacity to mitigate the vanishing gradient problem in standard RNNs, memory cells that allow for information storage and erasure, and gating mechanisms (input, forget, and output gates) to regulate data flow. These networks employ activation functions to analyze incoming data and train using "Backpropagation through Time" (BPTT). LSTMs find applications in a wide range of domains, from language modelling and stock price prediction to speech recognition and image captioning. Variants like Bidirectional LSTMs and simpler Gated Recurrent Unit (GRU) networks have also been introduced. LSTMs have proven highly effective in modelling sequential data, making them a

crucial component in deep learning applications, particularly for tasks involving sequences and time-dependent information.

## ResNet

Residual Networks, or ResNets [49], are a strong and innovative kind of deep neural network design that has had a significant influence on the disciplines of computer vision and deep learning since their inception in 2015 by Kaiming He et al. They were developed to overcome the difficulty of training extremely deep neural networks by addressing the vanishing gradient problem, which commonly impedes deep network training. The "residual block," which comprises two key routes, is the essential innovation of ResNets. The identity path reflects the original input and transfers it directly to the output, while the residual path applies a sequence of convolutional layers and non-linear activations to the input. Skip connections, also known as shortcut connections, allow gradients to flow more easily during training, allowing for the training of extremely deep networks with hundreds or thousands of layers without performance deterioration. ResNets have excelled in a variety of picture classification tasks, most notably in the ImageNet Large Scale Visual Recognition Challenge. Because of their effectiveness in lowering error rates, they are a popular choice for many image-related applications. ResNets are also widely employed in transfer learning, in which pre-trained ResNet models are fine-tuned for specific image recognition tasks with little data. Over the years, several adaptations and improvements to the original ResNet design, such as ResNetV2 and Wide ResNets, have been produced, significantly enhancing both performance and efficiency. ResNets, created for image classification, have found uses in other fields such as natural language processing and speech recognition. Residual Networks have had a tremendous influence on the area of deep learning, becoming a standard design for numerous computer vision applications, allowing the training of extraordinarily deep networks while preserving good accuracy and generalization capabilities.

## EfficientNetB4

EfficientNetB4 [50] is a member of the EfficientNet family of neural networks, known for their exceptional performance in image classification tasks while remaining computationally efficient. It strikes a balance between model size, computational requirements, and accuracy. EfficientNetB4 employs a systematic approach to scale neural network architectures, achieving an ideal balance of depth, width, and resolution through compound scaling. It uses depth-wise separable convolutions and squeeze-and-excite blocks to enhance efficiency and feature capture. This model demonstrates top-tier accuracy on benchmarks like ImageNet while being computationally efficient. EfficientNetB4 is widely used for transfer learning, where pre-trained models on large datasets are fine-tuned for specific image classification tasks with minimal data. Its efficiency and performance have made EfficientNet models, including B4, popular choices in various computer vision applications. It's crucial to select the appropriate model size, such as B4, based on the specific task's computational and accuracy requirements. EfficientNetB4 showcases an innovative approach to creating efficient yet high-performing convolutional neural networks, making it a valuable option for image classification and transfer learning.

## InceptionV3

InceptionV3 [51] is a CNN model widely used for image recognition tasks. It achieves high accuracy and features numerous convolutional, pooling, and activation layers. The architecture incorporates inception modules, enabling the network to learn distinct feature maps at

different scales. Batch normalization and factorized 1x1 convolutions are used to reduce parameters and improve training efficiency. While versatile for various tasks and datasets, it can be computationally intensive and memory-consuming.

## Xception

Xception [52], short for "extreme inception," is a deep CNN architecture proposed by François Chollet in 2017. It extends Inception's ideas by using depthwise separable convolutions, which are more efficient. Depthwise separable convolutions consist of depthwise and pointwise convolutions, reducing computational complexity. Xception is known for its deep architecture, which enables it to learn complex features, and it excels in image classification accuracy.

## MobileNet

MobileNet [53] is designed for embedded devices with limited processing capabilities. It balances accuracy and model size efficiently. The key innovation is the use of depthwise separable convolutions, which divide convolutions into depthwise and pointwise stages, significantly reducing computational costs and model size. This division drastically reduces computational demands and model size while maintaining reasonable accuracy levels.

MobileNet's efficiency is based on the separation of spatial convolutions (depthwise convolutions) from feature mixing (pointwise convolutions). This modular design allows MobileNet to efficiently learn and process information across layers while drastically reducing computing burden when compared to standard convolutional networks. Notably, MobileNet has progressed through several versions, including MobileNetV1, V2, and V3, with each iteration bringing improvements in speed and efficiency. These versions have optimised the design, utilising advances in deep learning techniques to expand its capabilities. MobileNet is widely used in mobile and embedded systems, playing an important role in tasks such as object identification, picture classification, semantic segmentation, and other computer vision-related applications. Its flexibility to resource-constrained devices, as well as its ability to retain competitive accuracy, makes it an ideal candidate for scenarios that need efficient yet strong neural network designs. MobileNet's use of depthwise separable convolutions, together with its growth through many iterations, highlights its importance in providing efficient and accurate neural network processing for mobile and embedded systems.

## Evaluation parameters

This study utilizes multiple evaluation criteria, such as accuracy, F1 score, recall, and precision, to gauge the effectiveness of transfer learning models. Furthermore, the research makes use of confusion matrices to assess the performance of these algorithms. A confusion matrix, also known as an error matrix, is a tabular representation commonly used to illustrate the classifier's performance on test data, offering a visual representation of algorithm performance.

A "True positive (TP)" refers to instances in which the model made an accurate prediction for the positive class, while "True negative (TN)" signifies cases where the model correctly predicted the negative class. Conversely, "False positive (FP)" corresponds to situations where the model made an incorrect prediction for the positive class when the actual class was negative. Likewise, "False negative (FN)" denotes instances where the model inaccurately predicted the negative class when the true class was positive.

The model's overall prediction accuracy is determined by evaluating the ratio of correct predictions to the entire dataset's total instances. This accuracy metric can be computed

through the following formula:

$$Accuracy = \frac{TP + TN}{TP + TN + FP + FN}$$ (1)

Precision serves as a metric that gauges the proportion of positive instances that were accurately predicted out of all the instances that the model identified as positive. Its central goal is to reduce false positives, providing insight into the model's capacity for correctly identifying positive cases. Precision is determined through the following formula:

$$Precision = \frac{TP}{TP + FP}$$ (2)

Recall, which is also referred to as the true positive rate or sensitivity, evaluates the proportion of positive instances that were correctly predicted with the total number of actual positive instances within the dataset. It quantifies the model's effectiveness in accurately capturing positive cases. Recall is computed using the following formula:

$$Recall/Sensitivity/TPR = \frac{TP}{TP + FN}$$ (3)

The F1 score represents the harmonic average of precision and recall, offering a well-balanced assessment of the model's comprehensive performance by simultaneously accounting for both precision and recall. Its computation involves the following formula:

$$F1score = 2 \times \frac{Precision \times Recall}{Precision + Recall}$$ (4)

## Results and discussion

The open-source TensorFlow and Keras libraries were used in this work to create the pre-trained models. The Python programming language was used in conjunction with the Anaconda platform to analyse traffic accident severity using transfer learning algorithms. A Dell Poweredge T430 server with a GPU was used to handle the dataset's computing demands. This server has eight cores, sixteen logical processors, and 32GB of RAM. The paper proposes using transfer learning techniques to handle the challenge of predicting traffic accidents. Various scientific approaches will be used to assess the efficacy and importance of the suggested methodology.

### Results of deep learning models for traffic severity prediction

Table 3 presents a comparative evaluation of the performance of three deep learning models_MLP, CNN, and LSTM_ in the context of traffic accident severity detection. Results reveal that CNN outperformed other deep learning models in terms of evaluation measures and achieved 89.37% accuracy, 86.63% precision, 88.67% recall, and 87.19% F1 score. It is followed by MLP which attained 87.27% accuracy, 82.29% precision, 83.61% recall, and 82.58% F1

**Table 3. Results of deep learning models for traffic accident severity detection.**

| Models | Accuracy | Precision | Recall | F1 score |
|---|---|---|---|---|
| MLP | 87.27 | 82.29 | 83.61 | 82.58 |
| CNN | 89.37 | 86.63 | 88.67 | 87.19 |
| LSTM | 81.27 | 80.24 | 83.29 | 82.67 |

**Table 4. Results of transfer learning models for traffic accident severity detection.**

| Models | Accuracy | Precision | Recall | F1 score |
|---|---|---|---|---|
| ResNET | 95.27 | 96.25 | 98.19 | 97.67 |
| EfficientNetB4 | 93.67 | 89.69 | 88.32 | 88.55 |
| InceptionV3 | 92.48 | 93.87 | 97.99 | 96.98 |
| Xception | 91.37 | 82.63 | 89.67 | 85.19 |
| MobileNet | 98.17 | 98.34 | 98.91 | 98.48 |

score. Deep learning model LSTM is the lowest performer model with 81.27% of accuracy score. The CNN model exhibited the most impressive results among other deep-learning models for predicting traffic accident severity.

## Results of transfer learning models for traffic severity prediction

Table 4 provides a detailed analysis of the performance of various transfer learning models employed in the context of traffic accident severity detection. It represents the performance of the different transfer learning models including ResNET, EfficientNetB4, InceptionV3, Xception, and MobileNet. The results demonstrate that MobileNet stands out as the top performer among the models, attaining the highest accuracy at 98.17%, alongside 98.34% precision, 98.91% recall, and 98.48% F1 score. In contrast, the Xception model ranks lower in terms of precision (82.63%) and F1 score (85.19%), indicating areas where it may require improvement. InceptionV3 and EfficientNetB4 have achieved 92.48% and 93.67% accuracy scores respectively. However, ResNET has shown the second highest results with 95.27% accuracy, 96.25% precision, 98.19% recall and 97.67% F1-score.

These findings are invaluable for researchers and practitioners seeking to leverage transfer learning for traffic accident severity detection, with MobileNet emerging as a particularly promising candidate for further exploration and deployment in real-world applications. The class-wise accuracy of all models is shown in Table 5.

## SHAP explanation

SHAP (SHapley Additive exPlanations) [54] is a popular technique used for explaining the predictions of machine learning models, including black-box models like deep neural networks used in transfer learning. The primary version of SHAP that is commonly used for explaining black-box models is called "Kernel SHAP." Due to its powerful processing and visualisation capabilities, researchers have been using it more frequently to study road safety [55, 56].

In this study, to uncover the significance of features in the black-box transfer learning model, Shapley values for each feature are calculated using the Python Shap library. SHAP

**Table 5. Class-wise results of transfer learning models for traffic accident severity detection.**

| Models | Fatal | Serious Injury | Minor Injury | Non-Injury |
|---|---|---|---|---|
| ResNET | 88 | 96 | 97 | 97 |
| EfficientNetB4 | 87 | 94 | 94 | 95 |
| InceptionV3 | 86 | 93 | 92 | 93 |
| Xception | 87 | 94 | 92 | 92 |
| MobileNet | 94 | 98 | 99 | 99 |

emphasises the significance of characteristics in forecasting water quality. While the relevance of the SHAP feature outweighs that of traditional approaches, it only provides limited extra insights when used alone.

The SHAP plot arranges features in descending order, indicating their importance, with high importance shown in red (towards the top) and low importance in blue (towards the bottom) along the Y-axis. The X-axis represents the impact of these features on the model output. Each point on the SHAP plot corresponds to a data point from the training dataset. When the X-axis value is to the left of 0, it indicates an observation that shifts the target value in a negative direction, while a value to the right of 0 shifts it in a positive direction. As depicted in Fig 2, the road category stands out as the most influential factor in the model's performance. In particular, higher road category values like 'Vehicle track' and 'Motorway' on the left side effect the model, which aligns with the fact that most accidents occur in rural and urban areas. In Fig 2, it is evident that drug consumption leads to more severe accidents. The Shapley value associated with the drug-related feature increases in tandem with drug consumption. In other words, as drug consumption levels rise, so do the probabilities of accidents and greater injury severity.

## Validation and generalization of the proposed approach

In addition to conducting validation using an alternative dataset, this study aimed to demonstrate the effectiveness of the proposed approach by conducting experiments on a distinct dataset known as "US accidents (2016-2021)" dataset [57] is a comprehensive repository of almost 2.8 million records recording traffic incidents that happened across 46 states in the United States between February 2016 and December 2021. This dataset provides extensive geographic coverage, including several locations across the country, as well as a five-year time range, making it useful for analysing regional and seasonal differences in accident patterns. The dataset, which has 47 variables, contains information on the causes that contributed to automobile accidents, including details on accident locations, timings, weather conditions, road conditions, and accident severity. Table 6 proving its effectiveness.

Time complexity in learning models is primarily concerned with the training phase, where it measures the time required to modify the model's parameters based on the input data. This complexity is determined by several factors, including the model's architectural complexity, the size of the training dataset, and the optimization approach used. Table 7 compares the training and testing times for the models utilized. Notably, the MobileNet model's computational time is surprisingly efficient, lasting just 200 seconds, significantly lower than the training timeframes of other transfer learning models used in this work. Impressively, this increased efficiency in training time does not compromise the model's accuracy, as it consistently outperforms individual models in terms of predictive accuracy.

To further assess the effectiveness of the proposed method, the K-fold cross-validation is incorporated as an additional step for performance evaluation. The results from the 5-fold cross-validation are presented in Table 8. These results demonstrate the superior performance of the proposed technique in terms of precision, F1 score, accuracy, and recall when compared to alternative models. Prominently, the low standard deviation values indicate consistent and stable performance across different folds, reinforcing the confidence in the trustworthiness and reliability of MobileNet.

## Discussion

The study's findings not only highlight the improved accuracy in predicting accident severity but also offer crucial insights into the significance of feature importance analysis, particularly concerning policy formulation and safety interventions.

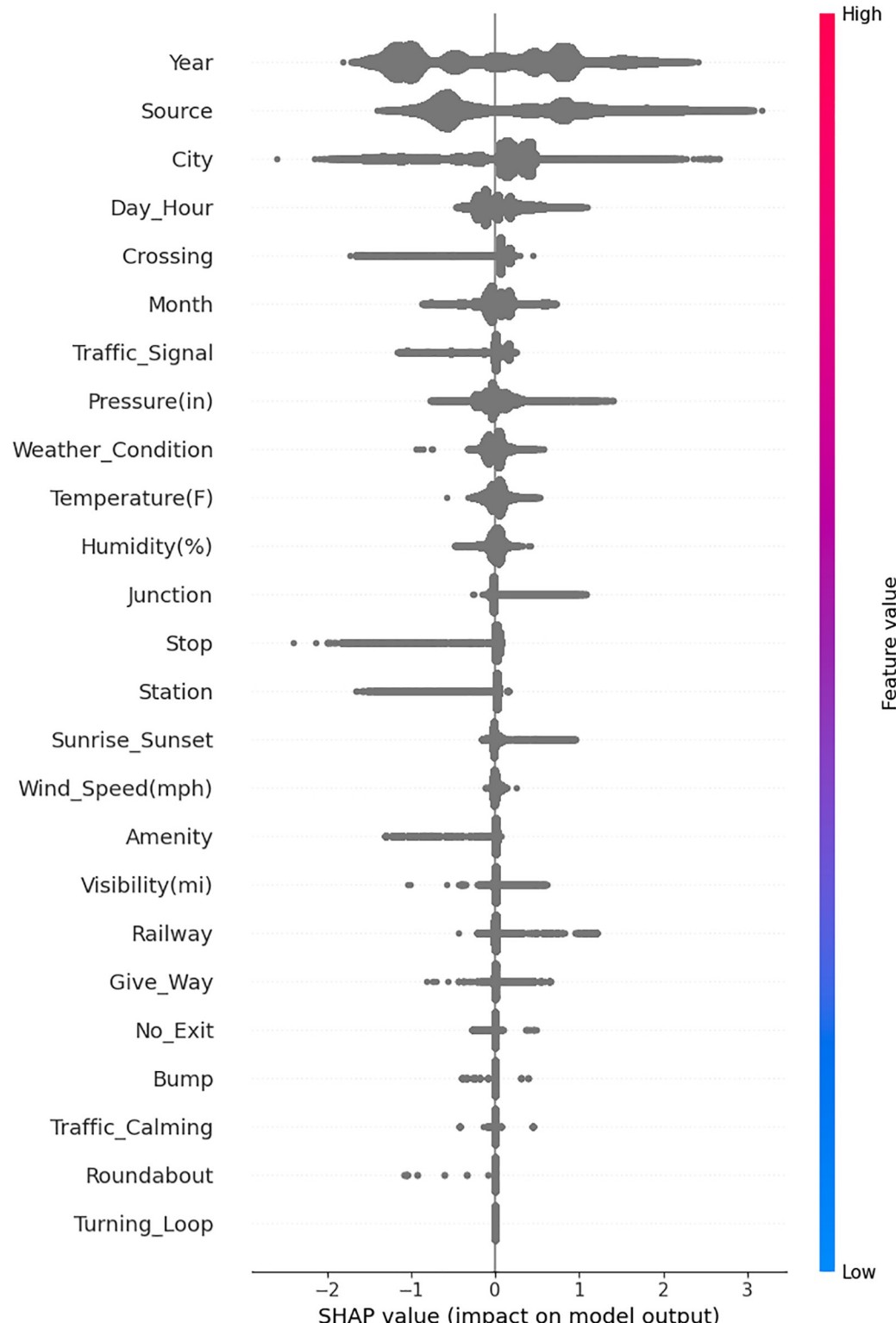

**Fig 2. Assessing the impact of features on the performance of MobileNet using SHAP.**

**Table 6. Result of MobileNet for traffic accident severity detection on US accidents.**

| Model | Accuracy | Precision | Recall | F1 score |
|---|---|---|---|---|
| MobileNet | 98.12 | 97.6 | 98.2 | 98 |

Feature importance analysis, exemplified through SHAP values, holds importance for stakeholders involved in transportation safety, policy-making, and law enforcement. By identifying influential features in accident severity prediction models, policymakers gain an understanding of the factors contributing most significantly to accidents. This understanding is pivotal for devising targeted interventions and formulating evidence-based policies to mitigate accident severity and frequency.

The interpretability facilitated by SHAP values enables transportation planners to prioritize interventions based on the most impactful features. For instance, if environmental factors or specific vehicle types consistently emerge as influential, policymakers can direct resources and interventions toward improving road infrastructure, enhancing vehicle safety standards, or implementing targeted awareness campaigns.

Moreover, the insights derived from feature importance analysis empower law enforcement organizations to optimize their strategies for traffic management and accident prevention. Identifying the key factors affecting severity aids in allocating resources efficiently, deploying enforcement measures where they are most needed, and devising preventive measures tailored to address the root causes of accidents. Additionally, the study's focus on MobileNet and the identification of influential features contribute directly to the development of more focused and effective actions aimed at reducing accidents. MobileNet's superior predictive accuracy, coupled with the understanding of influential features, presents an opportunity to devise proactive safety measures.

The broader implications extend beyond road safety. Accurate accident severity prediction aids in optimizing emergency response, insurance risk assessment, traffic management, and fleet safety. Moreover, its relevance in the realms of autonomous vehicles, public health research, urban planning, and smart city initiatives underscores its multifaceted significance. Furthermore, the application of feature importance analysis is not confined solely to road accidents. Its adaptability extends to aviation, maritime, and industrial safety, thereby enhancing safety measures across various domains and contributing to enhanced decision-making and accident prevention, ultimately saving lives and resources.

The feature importance analysis carried out in this study plays a vital role in informed policy decisions, targeted interventions, and overarching safety improvements, resonating across diverse sectors and aligning with the broader objective of ensuring safer transportation systems worldwide. Fig 3 provides the comparison of the transfer learning models and clearly shows the superiority of the proposed MobileNet.

**Table 7. Time complexity of transfer learning models (in seconds).**

| Model | Training Time | Testing Time |
|---|---|---|
| ResNET | 223s | 38s |
| EfficientNetB4 | 298s | 62s |
| InceptionV3 | 250s | 35s |
| Xception | 210s | 34s |
| MobileNet | 200s | 29s |

**Table 8. Findings of k-fold cross-validation.**

| MobileNet | Accuracy | Precision | Recall | F-score |
|---|---|---|---|---|
| Fold-1 | 98.73 | 99.83 | 98.73 | 99.26 |
| Fold-2 | 97.97 | 99.56 | 98.95 | 99.54 |
| Fold-3 | 98.79 | 99.97 | 99.94 | 99.86 |
| Fold-4 | 99.62 | 99.96 | 99.96 | 99.83 |
| Fold-5 | 99.47 | 99.95 | 99.93 | 99.94 |
| **AVG.** | **98.89** | **99.83** | **99.56** | **99.67** |

**Comparison with state-of-the-art.** Two studies are selected for comparison purposes. [27] applied the DNN model to detect injury severity and applied the layer-wise relevance propagation (LRP) method to explain the prediction outcomes. They used the Chinese traffic accident dataset in experiments. While LRP is a widely used technique, it has limitations such as sensitivity to hyperparameters and the choice of rules, and its explanations might lack the consistency provided by Shapley values. LRP primarily focuses on local explanations, attributing relevance to features for a specific input instance. The interpretation of LRP results can be more complex, and the choice of specific rules in LRP can impact the explanations.

While, [29] applied models (DT, NB, MLP, SVM, NN and ANN-MLP) and achieved 76.90% of accuracy on NHTSA-USA dataset. To extract significant features, they applied the Shapely decision plot. The use of the Shapely decision plot technique is commendable, but it is essential to note that different Shapley-based methods can vary in their interpretability and robustness. Shapely Decision Plots are sensitive to the sampling of instances used to compute

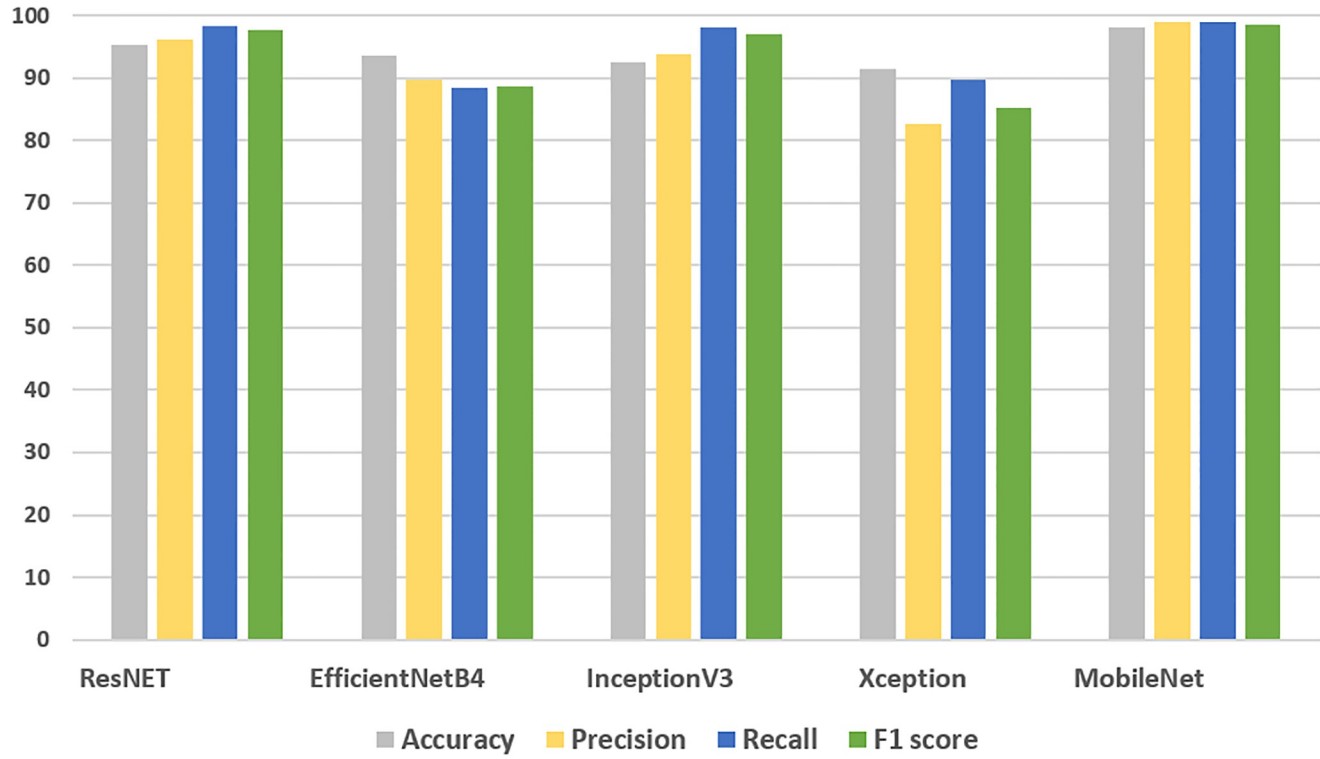

**Fig 3. Comparison of transfer learning models.**

Shapley values. If the sampled instances do not adequately represent the diversity of the dataset, the decision plot may not accurately reflect the true distribution of Shapley values. The NHTSA-USA dataset, while valuable for studying traffic accidents, has limitations. These include underreporting, inconsistent reporting, missing data, geographical and temporal biases, limited context, privacy concerns, data incompleteness, data imbalances, data collection bias, and changing data standards.

The proposed approach leverages transfer learning models on a US traffic dataset, enhancing generalizability by utilizing pre-trained models for improved performance and robustness. The introduction of the Shapely Beeswarm plot, based on Shapley values and cooperative game theory, is a noteworthy innovation, providing a theoretically sound and visually interpretable framework for explaining feature contributions across different predictions. Shapley values contribute to high interpretability and versatility, facilitating a comprehensive understanding of complex models and insights into the decision-making process. The demonstrated robustness, with an accuracy of 98.17%, indicates not only accurate predictions but also meaningful insights into the influential factors driving those predictions.

## Conclusion

Traffic accidents continue to pose a significant threat, resulting in loss of lives, injuries, and substantial disruptions on the roadways. Understanding the underlying factors that lead to these accidents is imperative for improving safety across transportation networks. The study leverages various transfer learning techniques and explains the most influential factors through the application of Shapley values. The research explored the prediction of accident severity using the models including Multilayer Perceptron (MLP), Convolutional Neural Network (CNN), Long Short-Term Memory (LSTM), Residual Networks (ResNet), EfficientNetB4, InceptionV3, Extreme Inception (Xception), and MobileNet. Among these models, the MobileNet stood out with the highest accuracy of 98.17%.

This knowledge provides a foundation for developing more effective measures to prevent accidents. In doing so, this research improves the accuracy of severity prediction and promotes the transparency, interpretability, and trustworthiness of learning models. This is essential for stakeholders and decision-makers seeking to take evidence-based actions to enhance road safety. Ultimately, the study's emphasis on transparency, interpretability, and the role of critical features serves as a cornerstone for informed decision-making in road safety measures, paving the way for a substantial reduction in the repercussions of traffic accidents on our highways.

## Author Contributions

**Conceptualization:** Omar Ibrahim Aboulola.

**Data curation:** Omar Ibrahim Aboulola.

**Formal analysis:** Omar Ibrahim Aboulola.

**Funding acquisition:** Omar Ibrahim Aboulola.

**Investigation:** Omar Ibrahim Aboulola.

**Methodology:** Omar Ibrahim Aboulola.

**Project administration:** Omar Ibrahim Aboulola.

**Resources:** Omar Ibrahim Aboulola.

**Software:** Omar Ibrahim Aboulola.

**Supervision:** Omar Ibrahim Aboulola.

**Validation:** Omar Ibrahim Aboulola.

**Visualization:** Omar Ibrahim Aboulola.

**Writing – original draft:** Omar Ibrahim Aboulola.

**Writing – review & editing:** Omar Ibrahim Aboulola.

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
