## [Decision Letter · Decision Letter 0]

13 Dec 2023

PONE-D-23-34929Improving Traffic Accident Severity Prediction Using MobileNet Transfer Learning Model and SHAP XAI TechniquePLOS ONE

Dear Dr. aboulola,

Thank you for submitting your manuscript to PLOS ONE. After careful consideration, we feel that it has merit but does not fully meet PLOS ONE’s publication criteria as it currently stands. Therefore, we invite you to submit a revised version of the manuscript that addresses the points raised during the review process.

We look forward to receiving your revised manuscript.

Kind regards,

Abel C.H. Chen

Academic Editor

PLOS ONE

Journal Requirements:

Reviewers' comments:

Reviewer's Responses to Questions

**Comments to the Author**

1. Is the manuscript technically sound, and do the data support the conclusions?

Reviewer #1: Partly

Reviewer #2: Yes

2. Has the statistical analysis been performed appropriately and rigorously? 

Reviewer #1: Yes

Reviewer #2: Yes

3. Have the authors made all data underlying the findings in their manuscript fully available?

Reviewer #1: Yes

Reviewer #2: Yes

4. Is the manuscript presented in an intelligible fashion and written in standard English?

Reviewer #1: Yes

Reviewer #2: Yes

5. Review Comments to the Author

Reviewer #1: The manuscript deals with prediction of crash severity using several machine learning techniques. The author has claimed the use of MobileNet transfer learning technique provides the best results. The author should address the following comments before publication:

The use of precise technical language: Avoid use of first person pronouns (I, we, us, etc.). Use commonly used technical terms, such as "lack" instead of "dearth".

Reference number [3] is not "recent", furthermore, it is not properly written in the reference list.

The full form of PART is missing.

The dataset is heavily biased towards non-fatal crashes. In that case, some balancing technique must be applied to have a valid model. IF not, then the model accuracy must be shown for each prediction class to show that the model is valid even without the use of balancing.

I believe SHAP analysis could be used for determining feature importance for any of the models. It has been done in the past. So, the only advantage of MobileNet, that could be claimed, is better accuracy. In terms of inherent explanatory power, models like CART are better.

The feature importance found through SHAP should be compared and discussed with other studies from literature.

Disucsion should be extended to show the policy and safety implications of feature importance analysis.

Why only two studies are selected for comparision of accuracy? Table 1 has a wider range of studies which could also be used for this purpose.

Figure 1 shows that data cleaning involved eliminating missing values, which is not mentioned in the text. How the missing values were eliminated? Did it involve eliminating the entire data record missing values?

Figure 2 needs a legend.

Importance features should be reiterated in the conclusion section.

Reviewer #2: Improving Traffic Accident Severity Prediction Using MobileNet Transfer Learning Model and SHAP XAI Technique

In this paper author predicts the severity of injuries in accidents, Multilayer Perceptron (MLP), Convolutional Neural Network (CNN), Long Short-Term Memory (LSTM), Residual Networks (ResNET), EfficientNetB4, InceptionV3, Extreme Inception (Xception), and MobileNet are employed. This paper may be considered for the publication after minor revision.

1. What is the contribution of this study, explain it

2. The whole paper is not justified by Grammarly.

3. Author should cite the related references of traffic prediction such as Bharti, Redhu, P. and Kumar, K., 2023. Short-term traffic flow prediction based on optimized deep learning neural network: PSO-Bi-LSTM. Physica A: Statistical Mechanics and its Applications, 625, p.129001.

Yadav, Sunita, and Poonam Redhu. "Driver’s attention effect in car-following model with passing under V2V environment." Nonlinear Dynamics (2023): 1-17.

4. The quality of the figures is very low.

5. Give more details about MobileNet

6. What is Shapley values.

7. Check Ref. [3]

6. PLOS authors have the option to publish the peer review history of their article (what does this mean?). If published, this will include your full peer review and any attached files.

Reviewer #1: No

Reviewer #2: No

---

## [Author Response · Author response to Decision Letter 0]

26 Jan 2024

I have added separate PDF file to address reviewer comments.

---

## [Decision Letter · Decision Letter 1]

4 Mar 2024

Improving Traffic Accident Severity Prediction Using MobileNet Transfer Learning Model and SHAP XAI Technique

PONE-D-23-34929R1

Dear Dr. aboulola,

We’re pleased to inform you that your manuscript has been judged scientifically suitable for publication and will be formally accepted for publication once it meets all outstanding technical requirements.

Kind regards,

Abel C.H. Chen

Academic Editor

PLOS ONE

Additional Editor Comments (optional):

Reviewers' comments:

Reviewer's Responses to Questions

**Comments to the Author**

1. If the authors have adequately addressed your comments raised in a previous round of review and you feel that this manuscript is now acceptable for publication, you may indicate that here to bypass the “Comments to the Author” section, enter your conflict of interest statement in the “Confidential to Editor” section, and submit your "Accept" recommendation.

Reviewer #1: All comments have been addressed

2. Is the manuscript technically sound, and do the data support the conclusions?

Reviewer #1: Yes

3. Has the statistical analysis been performed appropriately and rigorously? 

Reviewer #1: Yes

4. Have the authors made all data underlying the findings in their manuscript fully available?

Reviewer #1: No

5. Is the manuscript presented in an intelligible fashion and written in standard English?

Reviewer #1: Yes

6. Review Comments to the Author

Reviewer #1: (No Response)

7. PLOS authors have the option to publish the peer review history of their article (what does this mean?). If published, this will include your full peer review and any attached files.

Reviewer #1: No

---

## [Editor Report · Acceptance letter]

28 Mar 2024

PONE-D-23-34929R1 

PLOS ONE

Dear Dr. aboulola, 

I'm pleased to inform you that your manuscript has been deemed suitable for publication in PLOS ONE. Congratulations! Your manuscript is now being handed over to our production team.

Kind regards, 

on behalf of

Dr. Abel C.H. Chen 

Academic Editor

PLOS ONE